# Role of PPAR Receptor and Ligands in the Pathogenesis and Therapy of Hematologic Malignancies

**Jian Wu \*** **, Min Zhang and Allison Faircloth**

Division of Hematologic Malignancies and Cellular Therapy, Department of Medicine, Duke University Medical Center, Durham, NC 27710, USA; min.zhang116@duke.edu (M.Z.); allison.faircloth@duke.edu (A.F.)
\* Correspondence: jw731@duke.edu; Tel.: +1-(919)-381-8650

**Abstract:** The Peroxisome proliferator-activated receptors (PPARs) play vital roles in regulating cellular differentiation, proliferation, and caspase-mediated cell death pathways. They are regarded as promising targets for anti-tumor drug development, particularly for multiple myeloma (MM) and different hematological malignances. Several early section clinical trials are conducted to measure the clinical practicableness of PPAR agonists, notably PPARα and PPARγ agonists, against various cancers. A spread of studies has investigated PPARs expression in metabolic regulation. Furthermore, it has been suggested that careful designing of partial agonists for PPARs may show improvement with side effects and increase the therapeutic value. This review summarizes the organic chemistry and metabolic actions of PPARs, and the therapeutic potential of their agonists underneath clinical development. It investigates therapeutic agents for hematologic malignancies.

**Keywords:** hematologic malignancy; PPARs; agonist; ligands; metabolism

## 1. Introduction

The number of newly diagnosed cancer cases is on the rise, with 18.1 million instances diagnosed in 2018. Aside from cancer and diabetes, cardiovascular system disease is one of the most common and complex diseases currently having a significant impact on global health issues leading to death. Diabetes and cardiovascular system disease have a complicated and multifaceted pathophysiological relationship. Clinicians can benefit from a better understanding of the diseases' mechanisms to treat and prevent them [1]. Besides the increased risk of heart disease in diabetic people, scientists are also looking for a link between diabetes and cancer.

Diabetes is thought to increase the incidence of certain malignancies, which may be attributed to hyperinsulinemia [2]. Diabetic people are at a much greater risk for several forms of cancer. Epidemiologic research suggests that type 2 diabetes and cancer have many risk factors in common. Only a handful of the drugs used to treat hyperglycemia are linked to a high or low risk of cancer [2]. Many forms of cancer have been linked to type 2 diabetes, and it has been claimed that treatments for type 2 diabetes may affect cancer cells directly or indirectly [3].

Recent advances in molecular and cell biology have identified specific molecular targets for cancer therapy. Nuclear receptor superfamily members are an example. Estrogen receptors (ERs), retinoic acid receptors (RARs), retinoid X receptors (RXRs), and vitamin D receptors (VDR) are all members of this family. Because the aforementioned nuclear receptors are all transcription factors that affect the expression of several genes involved in carcinogenesis, there is a solid mechanistic basis for this targeting [4].

The role of peroxisome proliferator-activated receptors (PPARs) in numerous chronic diseases such as diabetes, cancer, and atherosclerosis inflammation, is well established [5]. PPARs are the superfamily of nuclear hormone that controls the genes involved in cell differentiation. The PPARs are expressed in the cardiovascular and their ligands have

shown a significant role in various cardiovascular risk factors in clinical and preclinical studies [6]. PPARs are ligand-activated transcription factors that belong to the nuclear receptor superfamily and are divided into three subtypes, PPARα, PPARβ/δ, and PPARγ [7]. PPARα is substantially expressed in brown adipose tissue and liver, whereas PPARγ is primarily expressed in adipose tissue. PPARβ/δ is mainly expressed in the gut, kidney, and heart [8]. Endogenous ligands for PPARs are fatty acids, triglycerides, prostaglandins, prostaglandins, and probably retinoic acid. Although various binding sites for PPARs in target genes have been reported, they share, in general, as a response element, a direct repeat of the sequence AGGTCA, spaced by a single nucleotide, which was originally identified for PPARα [9]; thus, if more than one of the receptors is expressed in a particular cell type, one could expect cross talk in response to endogenous or pan-PPAR pharmacological agonists. Specific agonists for PPARα are used classically for the treatment of dyslipidemia, and agonists for PPARγ are insulin sensitizers to treat patients with type II diabetes. Currently, no PPARβ/δ activators or antagonists are in official clinical use. A recent review summarized novel development regarding patients for PPAR modulators and possible novel clinical indications [10]. Clinical evidence, toxicological aspects, and side effects of the use of PPAR modulators have been reviewed recently [11,12]. All PPARs are involved in lipid and carbohydrate metabolism, homeostasis, cell proliferation and differentiation, inflammation, and cancer [13]. Increasing interest focuses on the potential implication of PPARs in cancer.

PPARα enhances lipid oxidation by facilitating metabolic remodeling, and their dysregulation contributes to metabolic disorders and liver disease [14]. PPARγ has mainly been studied as a key regulator of adipocyte differentiation, but it has also been studied for its function in hematological malignancies [15]. Ligand-activated PPARα heterodimerizes with RXRs (Retinoid X receptors), resulting in the binding of the peroxisome-proliferator response element (PPRE), which regulates target gene expression in inflammation, cancer, atherosclerosis, diabetes, and obesity [16]. This review aims to investigate the role of PPAR agonists in hematological cancers.

## 2. Background—Molecular Structure of PPARs

The PPARα gene is located on the human chromosome 22 (locus 22q12-q13.1) and consists of eight exons that code for the PPARα protein [17]. Human PPARβ/δ has been allocated to chromosome 6, at position 6p21.1-p21.2.5. It consists of nine exons, whereas PPARγ is found on human chromosome 3p25.2, spans a 100 kb region, and is structured into nine exons [18,19]. An N-terminal A/B domain, a DNA-binding C domain, a D domain, and a C-terminal ligand-binding domain (E/F domain) make up the modular structure of PPARs [20]. The target gene promoter region binds to the PPRE, which is attached to the C domain of PPARs. Ligands bind to the E/F domain, activating the targeted sequential expression of gene receptors. Upon ligand attachment to PPAR in its binding region, the translocation of PPAR occurs to the nucleus, producing a heterodimer with another nuclear retinoid X receptor. These PPARs then bind with the PPRE of the target gene [21]. These PPARs are involved in genes that regulate lipid metabolism and energy homeostasis, inflammatory modulation, cell proliferation, apoptosis, tumor progression, and spreading [22]. Furthermore, nuclear PPARs form heterodimers with retinoid X receptors (RXRs), and the balance of PPARs determines cellular behavior to retinol signaling [23]. In addition, PPAR/RXR heterodimers can also stimulate transcription due to ligand-dependent activation of PPAR or RXR. The relative level of co-factor expression is a vital determinant of the specificity of the physiological responses to PPAR or RXR agonists (Figure 1) [24].

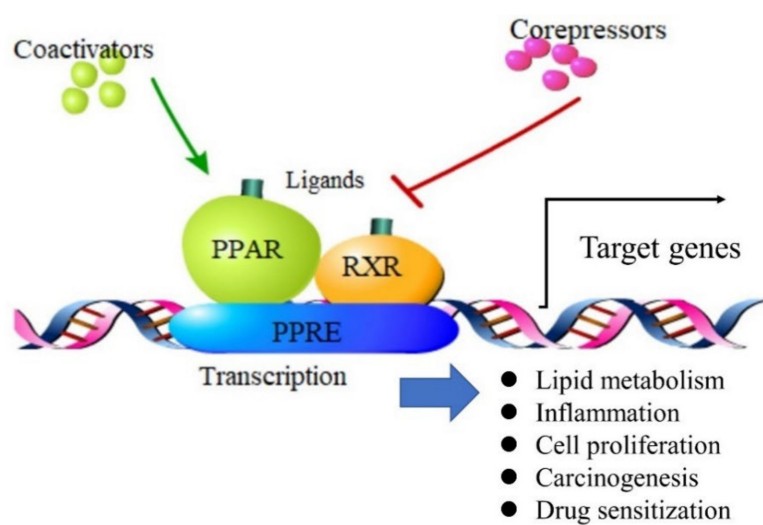

**Figure 1.** Interaction between PPARs and PPRE. The figure shows PPAR structure and related coactivator, and corepressor molecules involved in activation (green arrow) and repression (red arrow) mechanisms. The activation signaling of PPAR-RXR heterodimer and PPRE allows the expression modulation of target genes to affect various biological behavior.

Ligand binding causes conformational changes that allow coactivator or corepressor complexes to interact. The binding of cognate lipid ligands, heterodimerization with the retinoid-X receptor (RXR), engagement with a variety of transcriptional activators, and binding of the complex with the PPREs in the promoter region of the target gene are all required for the total transcriptional activity of PPARs.

It has been indicated that the ligand-binding domains share some standard features: (1) composed of 12 $\alpha$-helices arranged in an antiparallel helix sandwich and a four-stranded antiparallel $\beta$ sheet; (2) Y-shaped hydrophobic ligand-binding pocket with a volume of ~1300 cubic angstroms; (3) a C-terminal helix showing widely conformational variations in different crystals and playing essential roles in the activation of PPAR receptors [25,26].

## 3. PPARs Deletion Studies

Inactive PPARs have recently been linked to numerous areas. The early study revealed that macrophage-specific deletion of PPARγ results in decreased alternatively activated macrophages (AAMacs) in white adipose tissue (WAT) [27,28]. By regulating renal organic transporters MATE-1 and OCT-2, PPAR deletion reduces cisplatin nephrotoxicity [29]. In mouse myeloid cells, knocking down PPAR γ in mouse myeloid cells led to cardiac hypertrophy and increased myocardial infarct size [30]. Studies in mice with macrophage-specific deletion of PPARγ have shown that PPARγ is required for the maturation of AAMacs [31]. The effects of PPARβ/δ selective agonists on endometrial cancer cells apoptosis and gene expression are eliminated while PPARβ/δ was knockdown [32]. Using a pharmacological inhibitor or siRNA transfection to knock down or reduce PPAR expression during adipocyte development limits lipid accumulation [33]. In mice with a PPAR gene knockout, inflammation reaction to leukotriene B4 can be prolonged [34]. Radiation-induced apoptosis in the mouse kidney is inhibited by knocking down PPARα, which activates NF-KB and increases the production of IAPs [35]. These findings link PPARs expression to carcinogenic activity, emphasize their significance as tumor biology regulators, and justify the creation of PPAR antagonists or para-antagonists for clinical use.

## 4. Role of PPARs in Different Cancers

### 4.1. PPARs in Hematology Malignancies

Each member of the PPAR family has a particular role in different forms of biological activity: PPARα maintains energy homeostasis, PPARγ raised enhances glucose metabolism, and PPARβ/δ activation leads to fatty acid metabolism; therefore, the PPAR family is implicated in various disorders, including cardiovascular system disease, kidney disease, non-alcoholic fatty liver disease, and cancer.

#### 4.1.1. Chronic Lymphocytic Leukemia (CLL)

A recent study discovered that PPARα is expressed by circulating chronic lymphocytic leukemia (CLL) cells and is strongly linked to advanced-stage disease [36]. CLL may then depend more on PPARα-regulated fatty acid oxidation than aggressive lymphomas and acute leukemias, which employ aerobic glycolysis as a metabolic strategy. Rich acid oxidation rates in CLL cells can reach levels of more traditional fat-burning cells such as myocytes [37]. Unlike myocytes or hepatocytes [38], PPARα does not exhibit a significant daily rhythm in CLL cells, suggesting their dependence on fatty acid oxidation is unrelated to diet [36]. Perhaps PPARα expression in CLL cells reflects an origin in B1 B cells, which characteristically express high levels of this nuclear receptor [39,40].

Transgenic expression of PPAR increased immunosuppressive factors and tolerance to metabolic and cytotoxic stresses in CD5(+) Daudi cells [36]; however, significant downregulation of PPAR expression was linked to immunogenic death of developing CLL cells [41]. PPAR levels have been observed to be increased in CLL in numerous studies [42,43]. PPARα expression appears to be associated with more aggressive forms of CLL. CLL may then be more dependent on PPARα-regulated fatty acid oxidation than aggressive lymphomas and acute leukemias, which employ aerobic glycolysis as a metabolic strategy [44]. Above all, these findings suggest that inhibiting PPAR gene expression could be a potential CLL therapy strategy.

A substantial amount of research has been performed on PPARβ/δ in CLL cells. CLL cells had a higher level of PPARβ/δ expression than normal lymphocytes and other hematologic malignancies [45]. In stressful conditions such as depleted tissue culture medium, low extracellular glucose, hypoxia, and exposure to cytotoxic medicines, transgenic expression of PPARβ/δ enhanced the proliferation and survival of CD5+ Daudi cells and primary CLL cells [46]. Furthermore, in B lymphoma cell lines and primary CLL cells, PPARβ/δ was reported to promote Janus kinase (JAK)-mediated phosphorylation of signal transducer and activator of transcription (STAT) protein [47]. Above all, these findings show that targeting PPARβ/δ in CLL treatment may be therapeutic.

A recent study discovered that in MC38-OT I cancer model, the tumor microenvironment produces immunological defective tumor-infiltrating DCs (TIDCs), which can suppress dendritic cells' anti-tumor regulatory function, and this impact is produced by PPARα activation [48]; however, an exciting study indicated that the PPARα agonist Fenofibrate could boost the ability of a T-cell-inducing melanoma vaccine to delay tumor progression [49], which appears to contradict the finding above. These findings suggested that PPARα plays a significant role in hematologic malignancies.

#### 4.1.2. Acute Myeloid Leukemia (AML)

Acute myeloid leukemia (AML) is a clinically heterogeneous disease, yet it is one of the most molecularly well-characterized cancers. Despite the significant advances in the identification of new targets in the treatment of AML, still, no appropriate protocol has been developed for patients.

PPARγ is a single nuclear hormone receptor in the ligand-activated transcriptional factor that is thought to be involved in the regulation of lipid metabolism and plays a crucial role in various cellular functions, including insulin sensitization, inflammatory responses, and apoptosis [50]. PPARγ has a hand in the propagation of the signaling, mainly through interacting with the well-known tumor suppressor protein PTEN. Collected

bloods from 30 AML diagnosed patients and 10 healthy individuals revealed that the expression of PPARγ had an upward trend in AML patients. At the same time, PTEN significantly displayed a decreased expression compared to the control group [51]. The expression level of PPARγwas also negatively correlated with PTEN. Furthermore, previous studies indicated that PPARγ and its associated signaling pathway have a profound role in regulating cell proliferation and inducing apoptotic signaling by altering the balance between the expression level of pro-and anti-apoptotic target genes [52]. PPARγ stimulated in U937 cells not only transitioned the leukemic cells from the blocked G1 phase of the cell cycle but also significantly increased the number of the cells that underwent apoptotic death. PPARγ ligands have been shown to induce differentiation, promote apoptosis, and suppress proliferation of multiple AML cell lines (HL-60, KG-1, Mono-MAC6, and THP-1), as well as primary cells from AML patients [51]. This unique characteristic fired an enthusiasm to evaluate whether the stimulation of PPARγ in cancer cells could be a promising approach to cancer treatment strategies.

### 4.1.3. Multiple Myeloma (MM)

Multiple myeloma (MM) is a plasma cell malignancy that accounts for 10% of all hematological malignancies. Current treatment strategies for patients with multiple myeloma often now include high-dose chemotherapy with stem cell transplantation as part of their initial treatment. Although this approach has improved treatment outcomes for patients with MM, few patients are cured by this approach. As such, new treatment approaches for patients with MM are needed.

PPAR expression and activity were drastically reduced in adipocytes generated from MM patients. In MM-associated adipocytes, PPAR modifies adipokines was also reduced [53]. Many studies have claimed that the PPARγ overexpression decreased MM cell proliferation and induced spontaneous apoptosis, even without an exogenous ligand [54]. These PPARγ overexpressing cells were dramatically more sensitive to PPARγ ligand-induced apoptosis than uninfected or LV-empty infected cells. Apoptosis was associated with the downregulation of anti-apoptotic proteins XIAP, Mcl-1, and induction of caspase-3 activity. Importantly, PPARγ overexpression-induced cell death was not abrogated by coincubation with bone marrow stromal cells (BMSCs), which are known to protect MM cells from apoptosis [54].

### 4.1.4. Diffuse Large B Cell Lymphoma (DLBCL)

Diffuse large B cell lymphoma (DLBCL) is the most common subtype of non-Hodgkin's lymphoma (NHL) and accounts for up to 35–40% of all cases. Although the R-CHOP-based chemotherapy regimen could improve the 5-year overall survival of DLBCL patients to approximately 60%, 30% of patients would relapse; therefore, finding other effective therapeutic targets and strategies is of great clinical significance.

Accumulating evidence indicated that curcumin could promote cell apoptosis by suppressing the NF-κB signaling pathway in Burkitt's lymphoma [55] and cutaneous T-cell lymphoma [56]. A recent study demonstrated that PPARγ was a critical factor inducing the effects of curcumin. Curcumin inhibited Akt/mTOR signaling pathway via up-regulating PPARγ expression. A 10μ Mcurcumin treatment could up-regulate PPARγ expression in a time-dependent manner in DLBCL cell lines. In addition, PPARγ antagonist could reverse the effects of curcumin on DLBCL cells. Above all, PI3K/Akt/mTOR pathway was downstream of PPARγ, and curcumin might exert an anti-lymphoma effect through up-regulation of PPARγ [57].

Aside from PPARγ, PPARα was also reported as an anti-tumor factor that could regulate angiogenesis and inflammation. A meta-analysis of prospective studies regarding the association of BMI with the incidence and mortality for malignant lymphoma, which meant BMI had a relationship with different types of blood tumors, including DLBCL. Thus, with a PPARα agonist and lipid-lowering drug, we could significantly suppress tumor growth [58].

Moreover, a study showed that the Sirt1 gene as a potential target for DLBCL treatment was related to the PPAR signaling pathway, and when the PPAR pathway was blocked with a mitochondrial energy inhibitor, named "tigecycline", DLBCL drug (Adriamycin) resistance could be improved even with Sirt1 overexpressed [59].

### 4.2. PPARs in Solid Cancers

The PPAR family also plays an important role in solid tumor treatment. In addition, there are differing opinions about the therapeutic role of this family, but the penitential target is undoubted.

Some researchers have thought that PPARs are the reason that caused the tumor cell proliferation and anti-apoptosis. For example, one study revealed that a number of industrial chemicals and agents caused liver tumors by activating the PPARα, such as Key Evens (KEs) [60]. Davide Genini found that in non-small cell lung cancer, more apoptosis, less proliferation, and viability would happen during PPARβ/δ knockdown [61]. Xu's lab had a similar opinion, but for breast cancer [62].

However, there is the opposite view at the same time. According to Arezki's article, PPARα agonist Fenofibrate could improve the ability of a T-cell-inducing melanoma vaccine to delay tumor progression. In colon tumors, PPARβ/δ had lower expression in colon tumor tissue in both human or Apc$^{+/Min-FCCC}$ mice, compared with normal tissue [63]. Moreover, PPARγ, which expresses in adipose tissue, the large intestine, and spleen, also participates in the regulation of solid tumor development. Girnun et al. found that PPARγ could suppress tumorigenesis by downregulating the β-catenin, which was the marker of poor prognosis [64], and similar conclusions were made by Xu's group in breast cancer. They found that PPARγ could inhibit tumor cell proliferation in breast cancer through the ptgfp gene regulation, which made PPARγ become a potential target for treatment [65].

Currently, there is no consensus on the relationship between PPARs and tumors, many studies use different cell models, and tumor progression is determined by the interplay of cancer cell proliferation, angiogenesis, resisting cell death, and other multiple factors, so we cannot think of PPARs as a "good or bad" factor, but we are sure that PPARs are the "hallmarks of cancers" [66]. For future approaches using PPARs modulation for potential cancer therapy, collaborations between different laboratories and pathologists are urgently needed to define the exact expression patterns of PPARs in different types and stages of hematologic malignancies. The obtained data and co-administration of PPAR agonists with chemotherapy may provide new horizons for increasing the accountability and efficacy of cancer treatment.

### 5. PPARs Ligands

Given the critical role of PPARs in regulating malignant progression, different synthetic ligands based on subtypes of the PPAR family were studied. Other PPAR agonists and their current clinical statuses are illustrated in Table 1.

**Table 1.** PPARs agonists and their current status in clinical phase.

| Name of PPAR Agonist | Subtypes | Status |
|:---:|:---:|:---:|
| GW0742 | PPARβ/δ | Preclinical |
| L-165041 | PPARβ/δ | Preclinical |
| MA-0211 | PPARβ/δ | Phase I |
| KD-3010 | PPARβ/δ | Phase I |
| Oxeglitazar | Dual-PPARα/γ | Phase I |
| LY518674 | PPARα | Phase II |
| CHS-131 | PPARγ | Phase II |
| OMS 405 | PPARγ | Phase II |
| K111 | PPARα | Phase II |
| Efatutazone | PPARγ | Phase II |

NZT629, a highly selective PPAR antagonist, reduced agonist-induced transcription of PPAR-regulated genes, indicating target engagement [67,68]. NXT629 also causes CLL cells to apoptosis even in a protective microenvironment [42,69]. In vitro, NXT629 lowered the number of dividing leukemia cells [42]. In two xenograft mouse models of CLL, NXT629 reduced the amount of viable CLL cells [42,70]. Glucocorticoids and synthetic PPARβ/δ agonists increased PPARβ/δ expression while protecting Daudi and primary CLL cells from metabolic stress [46]. PPARβ/δ agonists enhanced plasmalemmal cholesterol and STAT phosphorylation, whereas PPARβ/δ deletion and chemical PPARβ/δ inhibitors lowered type 1 interferons (IFNs) [71,72].

Pioglitazone as an FDA-approved thiazolidinedione (TZD) drug for treating diabetes type II attracted tremendous attention due to its promising results in preclinical studies [73]. Saiki et al. reported that as compared to the cells with normal or lower expression of PPARγ, pioglitazone at a concentration ranging from 100 to 300 μM induced significant anti-survival and antiproliferative effects on PPARγ-expressing cancer cells [74]. Moreover, the results of several clinical trials on both acute myeloid leukemia (AML) and chronic myeloid leukemia (CML) patients indicated that pioglitazone at the dose of 45 mg/day is safe and well-tolerated [75,76]. Prost et al. delineated that when imatinib-resistant CML patients were treated with pioglitazone, the overall survival increased up to 4.7 years [77]. Given this, they extended their study to CD34[+] CML cells and concluded that pioglitazone might have the ability to eradicate the population of leukemic stem cells (LSCs)—a group of neoplastic cells that widely participate in the induction of chemo-resistance [77]. Ghadiany et al. have also administrated pioglitazone together with cytarabine and daunorubicin to newly diagnosed AML patients and indicated that the combination could increase the survival of the patients [75]. Based on these findings, we first evaluated the expression level of this nuclear receptor in the PBMNCs of patients with AML. In addition to managing blood glucose levels, pioglitazone has demonstrated significant efficacy in reducing cell survival in various human cancers [51]. Pioglitazone has been shown to effectively mitigate proliferation and cause apoptosis in breast and prostate cancer cells with enhanced PPARγ expression [78]. Furthermore, by suppressing the activation of signal transducers activator of transcription 3 (STAT3) and BIRC5 expression and increasing the levels of the apoptosis-inducing factor (AIF) levels in cancer cells, pioglitazone stimulation of PPARγ in cancer cells could induce apoptosis and cell cycle arrest [79]. Other solid tumors, such as colorectal cancer [80], Barrett's carcinoma [81], bladder cancer [82], and glioblastoma [83], have also shown that pioglitazone is a potential therapy option.

In five human MM cell lines (LP-1, RPMI 8226, OPM2, U266, and IM-9) and human bone marrow myeloma cells, treatment with pioglitazone, rosiglitazone, or 15-deoxy-δ 12,14-prostaglandin J2 (15d-PGJ2) decreased cell proliferation [84]. At a dosage of 50uM, 15d-PGJ2 generates a high rate of apoptosis in all cell lines [84]. Additionally, the combination of pioglitazone and arsenic trioxide (ATO) produced a signal that enhanced apoptotic cell death, most likely by elevating reactive oxygen species (ROS) and suppressing the PI3K pathway [85]. On the other hand, pioglitazone reduced U937 cell viability in vitro in a time and concentration-dependent manner [51]. These findings underline the role of PPARγ and its ligand pioglitazone in limiting leukemia cells' ability to sustain their survival and proliferative capability.

The natural PPAR agonist 15d-PGJ2 sensitized TRIAL-resistant cells to TRAIL, but this occurred independently of PPARγ signaling [86]. The synthetic oleanane triterpenoid CDDO (2-cyano-3,12-dioxoolean-1,9-dien-28-oic acid) can serve as a ligand for the PPARγ and has been shown to inhibit cell proliferation, and induce differentiation and apoptosis [87]. CDDO has been proven to cause apoptosis in hematologic malignancies in numerous studies [88].

Many anticancer treatment techniques interact with PPARγ ligands, which enhance their efficiency synergistically. A combination of PPARγ ligands and tyrosine kinase inhibitors (TKIs) in CML is the most striking example. In this situation, PPARγ activation

makes leukemic stem cells that had previously been resistant to treatment susceptible to targeted therapy [89].

Dual ligands for PPARα and PPARγ, such as C48 and TZD18, have been developed to improve the treatment of metabolic syndrome [90,91]. Interestingly, these dual ligands also possess anti-proliferation activities against various cancer cell lines with greater potency than conventional PPARγ ligands. A previous study demonstrated that TZD18 in leukemia cells might be independent of activation of PPARα and/or PPARγ [92]. C48 also seems to exert its growth inhibitory activity against CML cell lines via mechanisms independent of PPAR activation [93]. In addition, C48 and TZD18 also inhibited the growth of the imatinib-resistant CML cell lines K562/SR, KCL22/SR, and KU812/SR in a dose-dependent manner. These data suggested that the target of PPAR dual ligands in CML cells may not be the BCR/ABL protein, but, most likely, the downstream components of BCR/ABL signaling [93]. Since these dual PPARα/γ ligands exert their inhibitory effects on downstream targets of BCR/ABL signaling in CML cells, PPARα/γ dual ligands could enhance the efficiency of imatinib in BCR/ABL positive leukemia. This speculation has been confirmed by using TZD18 in BCR/ABL positive acute lymphocytic leukemia (ALL) and CML cells [91]. Other data also suggested that the combination of C48 and imatinib additively inhibited the growth of both imatinib-sensitive and imatinib-resistant CML cell lines as compared to either agent alone. In addition, the combination of imatinib and C48 resulted in a significantly higher level of apoptosis in K562 and KU823 cells than exposure to either agent alone [93].

## 6. Conclusions and Outlook

Many ligands have been created to target PPARs receptors and have helped researchers define the probable therapeutic target for drug development for nearly 20 years. PPAR modulators have gained a lot of attention because of the significant benefits of PPAR agonists in the treatment of metabolic illnesses.

For future approaches using PAPRs modulation for potential cancer therapy, collaborations between different laboratories and pathologists are urgently needed to determine the exact expression patterns of PPARs in various types and stages of hematologic malignancies. The findings and the co-administration of PPAR agonists with chemotherapy may open new avenues for improving cancer treatment accountability and efficacy.

**Author Contributions:** Conceptualization, J.W. and M.Z. Writing-original draft preparation, J.W.; writing- review and editing, M.Z. and A.F. All authors have read and agreed to the published version of the manuscript.

**Funding:** This research received no external funding.

**Institutional Review Board Statement:** Not applicable.

**Informed Consent Statement:** Not applicable.

**Conflicts of Interest:** The authors declare no conflict of interest.

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
