# Peer review of "Role of PPAR Receptor and Ligands in the Pathogenesis and Therapy of Hematologic Malignancies"

_hemato, doi:10.3390/hemato3030029_

Round 1

Reviewer 1 Report

The Authors addressed all comments. Thanks

Reviewer 2 Report

This manuscript has been much improved, though only one figure remained in it and no additional figure was added.

This manuscript is a resubmission of an earlier submission. The following is a list of the peer review reports and author responses from that submission.

Round 1

Reviewer 1 Report

In this review, the Authors have discussed some aspects of PPAR receptors in cancers. However, the review is not well organized and the arguments are mixed up between biological and clinical aspects. The title focuses on hematological malignancies, while a lot of non hematological conditions are present. Moreover, the Authors did not introduce hematological conditions, such as CML and CLL, and this lacks of clarity for non hematologists. Please include a brief description of each mentioned disease. Some parts can be shortened and combined, and better organized.

Reviewer 2 Report

This is an interesting and scientifically important paper that describes the roles of PPAR receptors and their ligands - first in general, later on with focal emphasis on the possible antineoplastic properties of this signal transduction pathway.

The manuscript is - in overall - well structured, however, only one figure is used to illustrate it. And even this figure is poorly readable due to the small letter font size used. A more extensive illustration of the complexities of the field is recommended. This is especially the case for the chapter that deals with the possible antitumor activity of this system, the main focus of the paper.

Unfortunately, the text is quite uneven: certain parts are very well readable while others have a more sloppy wording. Moreover, in some sentences important words seem to be missing. A thorough proofreading by a native English speaket is recommended. Just a few examples, (a non-exhaustive list):

  • Abbreviation of vitamin D-receptor is improperly placed within the sentence (VDR)
  • From cardiovascular system, the word system seems to be missing
  • the word mice is used as an adjective: mouse or murine should be used instead
  • These finding(s) - the s to express plural is missing
  • oxygen (104) - the citation is incorrectly placed

Nevertheless, I feel that the manuscript is a worthy and important one, after moderate rewriting

Round 2

Reviewer 1 Report

The Authors just addressed one point on the introduction of CML and CLL, while no reorganization has been done. Therefore, arguments are still mixed up between biological and clinical aspects, and also solid tumors are included (maybe change the title?). Some parts can be shortened and combined, and better organized.

Author Response

Point 1: The Authors just addressed one point on the introduction of CML and CLL, while no reorganization has been done. Therefore, arguments are still mixed up between biological and clinical aspects, and also solid tumors are included (maybe change the title?). Some parts can be shortened and combined, and better organized.

Response 1: We reorganize the manuscript and separate the PPARs function and the PPARs agonist. We also change the title to include the solid tumor. Meantime, some parts were shortened and combined.
